# Digital Pathology Workflow Implementation at IPATIMUP

**DOI:** 10.3390/diagnostics11112111

**Published:** 2021-11-15

**Authors:** Catarina Eloy, João Vale, Mónica Curado, António Polónia, Sofia Campelos, Ana Caramelo, Rui Sousa, Manuel Sobrinho-Simões

**Affiliations:** 1Pathology Laboratory, Institute of Molecular Pathology and Immunology, University of Porto, 4200-135 Porto, Portugal; jvale@ipatimup.pt (J.V.); mcurado@ipatimup.pt (M.C.); antoniopolonia@yahoo.com (A.P.); sofia.campelos@gmail.com (S.C.); acaramelo@ipatimup.pt (A.C.); rsousa@ipatimup.pt (R.S.); ssimoes@ipatimup.pt (M.S.-S.); 2i3S—Instituto de Investigação e Inovação em Saúde & Pathology Department of Medical Faculty, University of Porto, 4200-135 Porto, Portugal

**Keywords:** digital pathology, workflow, telepathology, implementation

## Abstract

The advantages of the digital methodology are well known. In this paper, we provide a detailed description of the process for the digital transformation of the pathology laboratory at IPATIMUP, the major modifications that operate throughout the processing pipeline, and the advantages of its implementation. The model of digital workflow implementation at IPATIMUP demonstrates that careful planning and adoption of simple measures related to time, space, and sample management can be adopted by any pathology laboratory to achieve higher quality and easy digital transformation.

## 1. Introduction

The Institute of Molecular Pathology and Immunology of the University of Porto (IPATIMUP) is a non-profit research institution with a pathology laboratory that is double accredited by the College of American Pathologists (CAP) and by NP EN ISO 15189 standards. It serves as a reference center for second opinions on difficult cases, biomarker identification, and training of pathologists and laboratory technicians. The experience with a telepathology project that started in 2013, and the wide use of scanned dark field images for fluorescent in situ hybridization tests, motivated the quest for digitization of the laboratory. Successful digital transformations of pathology workflow have been published in the literature [1,2,3]. The advantages of the digital methodology are well known and include time sparing workflows, as well as a reduction in costs [4,5].

The Food and Drug Administration (FDA) approval of the first scanning systems for primary diagnosis constitutes a relevant driver for the adoption of a digital workflow, representing general support of the regulatory institutions on the subject. The use of scanning systems other than those approved for clinical use by the regulatory institutions should be performed under strict surveillance by internal/external quality control programs [6].

In this paper, we describe the process for digital transformation of the pathology laboratory at IPATIMUP, including a detailed description of the modifications operated throughout the processing pipeline, as well as the advantages of its implementation.

## 2. Materials and Methods

The process for digital transformation of the laboratory started in 2016 when we decided to start preparing the staff and the respective structure. Pathologists and technicians underwent sessions of training and courses to understand the best way to start applying modifications to the laboratory, namely space and time, a new type of management, equipment renewal/acquisition, information technology infrastructure, and design of the validation of digital observation by the pathologist. The goal was to introduce whole slide images (WSIs) for diagnosis in all bright field tissue-related cases. Dark field fluorescent in situ hybridization (FISH) and immunofluorescence had already been achieved by a digital process since 2014 after the optimization of the D-Sight FLUO 2.0 scanner (Menarini Diagnostics^®^, Florence, Italy) for capture, matching of the fluorescent and haematoxylin and eosin (H&E) images, and semi-quantitative analysis, substituting an immunofluorescence microscope. At IPATIMUP, only routine cytology is left to be integrated in the digital workflow. The services provided by the pathology laboratory of IPATIMUP do not include, at the moment, autopsies or frozen sections.

We choose the Pannoramic^®^1000 (P1000) scanner (3DHISTECH Ltd.^®^, Budapest, Hungary) to obtain WSIs for primary and secondary diagnosis of all slides managed in the laboratory (100%), except for those of cytology, as mentioned above. Cytology slides that needed to undergo second revision in another institution or that were estimated to be consumed by molecular techniques were also scanned.

In July 2019, a P1000 scanner was installed in the center of the main laboratory surrounded by benches where specimen processing takes place. The scanning process, including quality control of the WSIs obtained was performed by trained technicians. All scanned slides were orderly incorporated in the file of the patient for microscopic observation after the functional integration of the scanner software with the laboratory information system (LIS) called SISPAT (JSalgado^®^, Porto, Portugal).

### 2.1. Digital Workflow

We describe the processing pipeline with emphasis on the major alterations introduced in the workflow of the pathology laboratory of IPATIMUP. For the successful implementation of these alterations, close interaction between technicians and pathologists was mandatory in order that the measures taken had no impact on the turn-a-round time or quality of the final product. Overall, there was an important investment in space contraction on the main laboratory area, since no important infrastructure interventions were done. The parallel benches were organized according to the flow of the sample, following a Lean approach, and allowing the insertion of a scanner station (Figure 1).

The scanner station was located in the confluent end of the histology and cytology lines, away from the paraffin-rich area (Figure 2). The location of the scanner within the main laboratorial area enabled a better communication process and fast management of samples. The disadvantages of having the scanner in the main laboratorial area were the increment in environmental noise produced by the instruments and the exposure of the scanner to potential particles produced during the entire process.

The same contraction exercise was applied in the management of time. Since the number of technicians was not increased and the scanning process imposed additional time spent on the technical side, an effort was made to reduce lost time, redundant tasks, or uncoordinated efforts, increasing the overall efficacy of the laboratory. Specific goals related to the time for production of stained and unstained full rack slides were established to better manage the occupation of the scanner station.

During the preparation period of the laboratory, and before the scanner acquisition, importantly, we implemented a sample tracking system based on the LIS which facilitated mobile control of time and operator’s intervention during the entire process. The tracking system was designed to use QR code readers at each station. Printed QR codes are part of the sample redundant identification in all phases of processing, from when the sample enters the institution until the report is signed out, including the physical and digital archives. This decision required the acquisition of computers or tablets for each workstation.

### 2.2. Sample Management and Macroscopic Examination

Good quality samples are easy to manage in the laboratory as compared with those with fixation problems that require additional time-consuming procedures to be suitable for diagnosis. To decrease the time required to manage problematic samples and to increase the quality of the image for diagnosis, an educational program was elaborated targeting nurses and physicians and highlighting the importance of controlling pre-analytic conditions. The administrative team was also trained to be able to identify problems with the packaging of samples in order to quickly promote their correct fixation by the technical team. The traffic of labeled samples with QR codes ran from the reception to the macroscopy room in scheduled batches and was performed to reduce people movements while keeping the macroscopy station as busy as possible.

After the QR code labeled samples were transferred to the macroscopy room, they were photographed using MacroPATH (Milestone Medical^®^, Bergamo, Italy), and fragments were collected to QR code printed cassettes (Figure 3). The photography system and cassette printer were connected with the LIS.

The size of the fragments collected was adjusted to the area of the slide that was captured by the scanner, away from the borders. At this station, the cassettes were organized immediately in the processor racks separating the exams associated with the fast scanner (usually small biopsies) from those with a prolonged scanner time (usually large surgical specimens). These two types of exams were kept separated during the subsequent histology processing so they could be managed easily at the scanner workstation. Prioritization of urgent samples was also done at this station.

The inking of specimens was always adjusted. The colors selected to ink surgical margins were those best identified by the scanner, providing a clear image during WSI observation. Cellblocks and breast cancer biopsies (rich in adipose tissue, nearly transparent mainly after immunohistochemical stain) were also inked before processing so that the cores and the pellets of cells were automatically detected by the scanner.

### 2.3. Processing, Embedding, Cutting, Staining, and Mounting

During the aforementioned steps of specimen handling at the laboratory, traceability and records from each station were kept in the file of the patient at the LIS. Records of reagent changes and equipment performances were kept, granting the identification of causes for poor quality products.

In addition to improving space, time, and sample flow, we organized the histology pipeline into a continuous production of slides to scan. Embedding was now performed according to priorities, taking in consideration that fragments must be placed close to each other and in the center of the paraffin block to decrease the scanner area and avoid placing tissue in the non-scanned limits of the slide.

We improved the cutting station process by introducing updated microtomes that allowed a stable thickness of the tissue for an uneventful image capture. The confirmation of the paraffin block entry at the cutting station with the QR code reader ordered the print of the respective labeled slide, reducing the transcription errors, accelerating the identification, and transferring the QR code ID to the slide that would be read by the scanner.

The staining and mounting process was fully automatic and operated with the Tissue-Tek Prisma^®^ Plus & Tissue-Tek Film^®^ (Sakura^®^, Nagano, Japan) integrated system, following an optimal protocol with daily reviewed reagents and contaminant controls, to obtain the best and stable staining observed in WSI. The selection of the staining and mounting equipment took into consideration the compatibility of the racks with those of the scanner; the scanner had been calibrated by the manufacturer according to the coverslip film used to adjust focus distance. Stained and mounted slides were dried in a 60 °C oven for 5 min to guarantee complete drying of the slides.

### 2.4. Scanning and Quality Control of WSIs

The glass slides were produced in racks, were orderly prioritized, and continuously arrived at the scanner station.

The scanner workstation consisted of a scanner and two computers. One was an Intel^®^ Xeon Gold 5120 @ 2.20 GHz (Intel^®^, Santa Clara, CA, USA) processor, 96 GB of memory, a 240 GB SSD disk for 64-bit OS, 960 GB SSD for SWAP and a 2TB mechanical disk for local storage that gathered the WSIs, converted them using the 3DHISTECH Slide Converter, and then stored the slides in the 3DHISTECH CaseCenter server located at the building data center. The 3DHISTECH Slide Converter compressed all the WSI files by 80%. The connection to this server was performed by a non-dedicated 1 GB network that served all infrastructure. The CaseCenter server had an Intel^®^ Xeon E3-1270 v6 (Intel^®^, Santa Clara, CA, USA) @ 3.80 GHz, 24 GB of RAM, 2x 240 GB SSD for 64-bit OS in RAID 1, and a 20TB volume of mechanical disk in RAID5. Through iSCSI, this server connected to the digital archive IBM FlashSystem 5000 with 220TB storage (that can scale up to 960TB to give extra volumes to the server) with distributed 6 RAID disk configuration. Another computer was used for WSI quality control operating in the patient files at the LIS.

At the scanner workstation, slides were transferred from the stainer racks to the scanner racks, according to the manufacturer’s instructions. The racks were introduced in the P1000 position that required less movement of the scanner operative arm. After the scanning process using a 20× adapted protocol (0.25 µm pixel size), the WSIs were automatically transferred to the patient’s file at the LIS and were available 30 s (average) after capture. Special protocols, such as those used in breast cancer biopsies and bright field in situ hybridization, used a 40× lens.

In the same station, all WSIs were opened by the technician and the WSI quality control process started. In each case, there was a verification of the matching of the identification, matching the number of fragments per slide in the WSI according to the photo of the slide captured by the scanner, and a verification of the focus and staining overall quality. If an irregularity was detected at this verification the technician, assigned for the quality control, recorded it at the LIS and ordered the return to the analytic phase where the error had occurred. In this situation, the original WSI was deleted to be substituted by the correct one. All WSIs used for diagnosis were archived and preserved for future consultation. If the case was ready for review by the pathologist, the technician released the file to enter the WSI in the diagnosis phase. The pathologist’s assignment plan was determined daily, prior to the embedding phase.

Slides generated in the setting of complementary techniques, including histochemical stains, immunohistochemical stains, and bright field in situ hybridization were prepared following the aforementioned standards and following specific scanning protocols adjusted for each type of technique. Immunohistochemical slides required, after the washing step, extra dehydration and prolonged diaphanization to avoid drying artifacts and residues in the respective WSIs.

The complimentary technique slides all always included, in addition to the sample, a set of positive and negative controls (2–5 tissue cores) specific for the technique used in the slide (Figure 4). The production of traceable and reliable tissue microarray control sets required the construction of a quality regulated tissue control bank.

### 2.5. WSI Review and Diagnosis

We targeted the environment at each pathologist’s office for modifications, with the purpose of creating comfort/ergonomics for the pathologist who reviewed cases using a monitor. A larger desk with space to accommodate two monitors was installed and organized to allow wide-ranging movements of the mouse for navigation. Light regulation of the environment required the installation of blackout shutters on the windows, to be used on demand.

The workstation of the pathologist included one Dell Precision Tower 3620 equipped with an Intel^®^ Core i7-6700 CPU (Intel^®^, Santa Clara, CA, USA) @ 3.40 GHz, 8 GB of RAM, ST500DM002-1SB10A ATA Disk with 466 GB and a NVIDIA QuADro M2000 (NVIDIA^®^, Santa Clara, CA, USA) with 4 GB. This workstation had two monitors one Sharp PN-K322BH (3840 × 2160 resolution in dots—QFHD, 32″) for slide analysis and one smaller monitor for regular tasks, i.e., a Dell (Dell^®^, Round Rock, TX, USA) P2417H (Full HD, 24″). The computer was connected to the LIS, CaseCenter, CaseViewer, and to other computers in the laboratory by a 1 GB network. Remote access to each workstation was available through a VPN connection that allowed the pathologist to work at a distance whenever it was required.

Management of all the information belonging to a case/patient was performed at the pathologist’s workstation using only the LIS, including access to all clinical data, previous and simultaneous exams, and respective WSIs, pre-analytical data, analytical data including macroscopic description and photographs, WSIs of the current case (H&E and complimentary techniques if available), microscopic description and diagnosis templated, codification system, quality evaluation form, and sign out area, in addition to all the relevant information regarding deviations from the regular laboratory workflow.

The validation of the digital WSI observation for clinical use was performed using the CAP guidelines applied to each pathologist [6].

## 3. Results

The pathology laboratory of IPATIMUP designed a digital transformation of the workflow that started in 2016 with the introduction of pre- and post-scanner modifications. The scanner was installed in July 2019, the software functional integration with the LIS was achieved during October 2019, the quality control program was adapted during November 2019, and the validation for clinical use lasted until July 2020. During this validation process, a hybrid workflow was maintained, providing both glass slides and respective WSIs to the pathologists. Since July 2020, 8 out of 14 pathologists have been receiving WSIs for primary diagnosis instead of glass slides. The remaining 6 pathologists are not using WSIs because they are reviewing only cytology cases (*n* = 1); they are part of the telepathology project that includes mainly tele-macroscopy and is managed by a different source and software (*n* = 2) or they are reporting an average of less than 10 cases per month (*n* = 3). If we consider those pathologists that could use, in fact, WSIs for diagnosis, only 3 out of 11 pathologists were missing (27.3%), representing a percentage of adhesion to WSI of 72.7%. The laboratory activity encompasses about 40,000 paraffin blocks and 60,000 slides per year reflecting the management of nearly 25,000 cases per year. These numbers do not include those cases received from other institutions for second opinion and biomarkers evaluation. The slides produced or arriving from an external source that configure histology, cellblocks, histochemistry, immunohistochemistry, in situ hybridization (both bright and dark field), and direct immunofluorescence are all scanned. Table 1 summarizes the WSI bright field production of the 8 months operating fully digitally (from July 2020 to February 2021).

The average number of slides scanned per day is 326 with a total of 57,418 slides generated in 8 months.

The reasons for rescanning slides are poor focus or incomplete scanning of the fragments and/or difficulties associated with uneven thickness of the tissue. The most frequent cause of scanner failure is the misprinted QR code, thus, leading to failure to scan sections placed in the lower limits of the slide. The automation of the mounting process with restricted human manipulation of slides (wearing gloves), together with the lack of glass coverslip corners misaligned with the slides, enables clean preparations that are easy to adjust to the scanner racks.

The cases requesting glass slides for diagnosis include those illustrating breast or prostate cancer biopsies presenting suboptimal material for nuclear evaluation, cases suspicious for amyloid deposition with the need of polarized light technique after Congo Red staining and, mostly, intrinsically difficult cases. The preventive maintenance of the scanner (single scanner) that occurred in February (Table 1) justifies the increment in the number of cases needing glass slides during this month. We have no records of slide breakages so far, nor scanner mal functions due to poor handling by the technicians.

The average size of slides and respective time for scanner concerning the type of specimen is summarized in Table 2. Cellblock slides have always two sections, an average of 1400 megabytes in size, and take an average of 100 s to scan. The time to scan a 1.5 × 1.5 cm tissue sample is 51 s.

Validation of all types of preparations by each pathologist using the digital pathology model was achieved and approved after over 95% concordance rates (using the microscopic observation at the optical microscope for comparison purposes).

As a result of the measures operated in the workflow, we obtained the following results:A 35% decrease in inadequate samples is recorded after the educational program targeting nurses and physicians to improve the quality of the pre-analytic conditions.Case assignment is facilitated as it is recorded at the LIS.Less than 24 h is needed from when a sample arrives at IPATIMUP until the respective (H&E) WSI is ready to review, allowing the establishment of a 48 h benchmark for turn-a-round time of all exams that do not need complementary techniques.The quality of the laboratory product was not affected by the digital workflow implementation according to the registries in the internal quality control program of the laboratory.The quality of the diagnosis produced in the laboratory was not affected by the digital workflow implementation, according to the results on the external quality control program.During the COVID-19 pandemic lockdown, the pathologists keep working either at home or at the laboratory using WSIs to diagnose and to share cases, and asking for a second opinion using digital tools to annotate diagnostic specific questions. Flexibility in scheduling reviews is facilitated by the remote access; pathologists continued quality control activities at a distance, by observing WSIs for validation of techniques; technicians were able to do the quality control of WSIs for diagnosis at a distance.Consultation of WSIs from previous or simultaneous exams from a patient is facilitated due to easy access to the digital image sparing time in retrieving glass slides from the physical archives.To archive glass slides becomes easier since the slides travel from the scanner station to the physical archives in the proper order.Costs with paper and printing were 25% reduced during the last year due to the transformation of paper records into digital ones, also offering the possibility of an ecological attitude welcomed by the team.Sharing cases with other institutions for secondary observation using a digital link for WSIs of patients in 108 cases/629 slides during the first 6 months represents faster and cheaper communication, which also prevents loss of material and glass slide damage.The digital workflow implementation brought new life to the research initiatives of the laboratory, as we had previously described [7].

## 4. Discussion

The digital transformation of the pathology laboratory at IPATIMUP is an example of successful implementation of the digital methodology for conducting pathology workflow at the tissue level (including cellblocks) for primary diagnosis. At IPATIMUP, only cytology is left to be integrated into the digital workflow. This is due to the very successful and intense production of smears after fine-needle aspiration that are difficult to manage in the WSI format. The WSIs of smears are time- and storage-consuming and usually do not reproduce the entire slide surface, leaving the limits of the slide left to be captured [8,9].

At IPATIMUP, we record a very low percentage of glass slide utilization (2.3% of cases) and the adoption of the WSI by the majority of pathologists (72.7%), indicating that pathologists trust the new methodology and understand its benefits. Pathologists must not be forced to accept this methodology change since it may compromise their diagnostic performance. The reasons for lack of acceptance may be related to expectations, habits, type of reported exams, and work conditions (speed of refresh, color calibration, and monitor quality) [2,10,11]. To prevent the lack of acceptance at our laboratory, we invested in creating comfort conditions at the office, individual participation in the validation process, laboratory informatic system (LIS)-centered operability, and diagnosis training under the new conditions during the extended hybrid workflow. Maintaining the possibility to revise the glass slides is advised since the sense of “no turning back” is avoided, and also because the reasons that triggered such requests may represent the need to optimize scanning protocols or situations in which technology needs to evolve (such as the use of polarized light for amyloid detection that is only available in some scanners) [3].

The technicians’ trust in the digital methodology is also relevant to keep the team motivated, something that is measured by the low volume of rescanning (average 1%) and the high classification attributed to the slides by the pathologists at IPATIMUP (average 98.3% are good). The confidence of the technician in the laborious process of digital transformation may be threatened if a prolonged hybrid workflow is maintained, preventing the immediate collection of the benefits inherent to the new methodology.

The implementation of the abovementioned measures in the pre-scanner process, namely those related to the high performance automatic stainer and coverslipper compatible with the scanner, are those that motivate the low volume of rescanning, as also reported in the literature [1]. Furthermore, the scanner was calibrated by the manufacturer according to the film used in the automatic mounting to adjust focus distance.

The results, herein presented, suggest that the pre- and post-scanner segments of the workflow adaptations are, at least, as important as the choice of the scanner and can be an important cause of implementation failure.

Indirectly, digital transformation stimulates an increment in quality control measures such as the tracking system, improving safety, and stimulating the creation of validation habits and risk-oriented thinking. Specific features of the digital methodology that are related to increased quality and safety include the possibility of rapidly sharing cases for a second opinion at a distance, reducing the distance between people in adverse situations such as those experienced during the COVID-19 pandemic [12], and the possibility of archiving WSIs representative of glass slides requested by other institutions or destroyed by molecular techniques [1].

The time and space contraction measures operated at IPATIMUP, very much inspired in the Lean approach, are mainly without cost, improve workflow efficacy, and are useful for digital and non-digital laboratories. The digital pathology model implemented at IPATIMUP demonstrates that the turn-a-round time can be maintained after the digital transformation with the same amount of technical and medical staff, provided the workflow is carefully optimized. Differently, the overnight scanning process adopted by other laboratories may not be compatible with the preservation of the present turn-a-round time and occurs, in most instances, during an unsupervised period [1]. The adoption of simple measures at IPATIMUP such as mounting with film, drying the slides, and careful transfer of slides between racks, helped to prevent bad functioning of the scanner, loss of time, and additional costs.

Time and resource control measures include the identification of bottleneck stations where samples may accumulate. In our laboratory, the bottleneck stations are those of the macroscopy and scanner. At the macroscopy room, acceleration of the descriptions and documentation could only be achieved by the addition of a new technician, a measure that for now is not yet cost effective. At the scanner station, the benefit of having two scanners with low capacity instead of one with large capacity has been defended by some authors [1] and can benefit the general workflow since this possibility also represents the existence of a backup during malfunctioning or preventive maintenance intervention.

The most frequent reason associated with refusal to implement a digital pathology workflow is cost related [3]. In the digital pathology model implemented at IPATIMUP, the requested acquisitions were distributed in time, with the most relevant being related to the automatic strainer and coverslip, the scanner, and the digital archive. In addition, saving costs were possible due to a reduction in post office trades, as well as prints and paper representing an ecological attitude. In the future, the use of image analysis algorithms and the possibility of operating in a scalable economy, will certainly imbalance the costs.

The digital archive is a hot topic related to digital pathology with both positive and negative opinions about a permanent archive of WSIs [13]. We agree that the smaller the archive the better it is to manage and, in our laboratory, with relatively low volume, we may sustain a privileged position to easily achieve the digital transformation. Again, in line with the simple adopted measures described above, to use validated 20× scanning protocols, the concentration of the fragments at the embedding station, image compression balance, and the substitution of poorly focused WSIs by high quality ones may prevent unnecessary archive consumption.

The model of digital workflow implementation at IPATIMUP demonstrates that careful planning and adoption of simple measures related to time, space, and sample management may be adopted by any pathology laboratory to achieve higher quality and easy digital transformation. Without digital transformation, pathology laboratories will not be able to benefit from the advantages provided by the WSIs, namely the application of computational pathology tools that are transforming the way we integrate molecular pathology and tissue morphology [14].

## Figures and Tables

**Figure 1 diagnostics-11-02111-f001:**
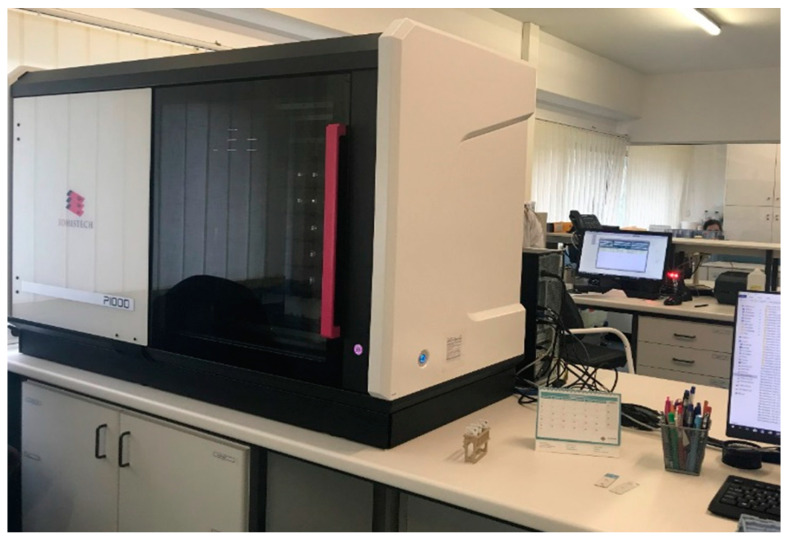
The scanner is located at the heart of the laboratory surrounded by benches where specimens processing takes place.

**Figure 2 diagnostics-11-02111-f002:**
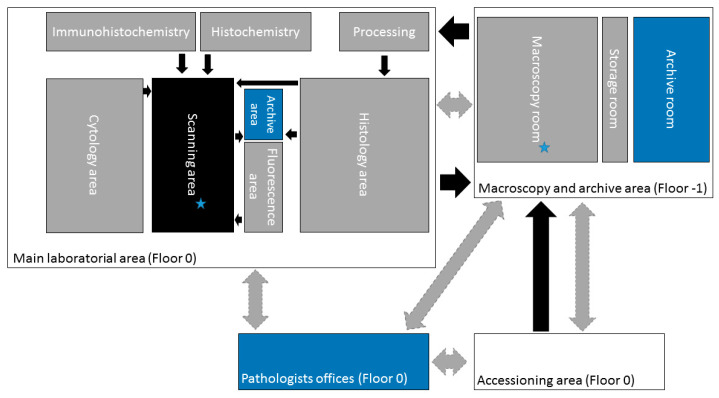
Schematic representation of the laboratory organization. The gray areas represent pre-scanner workstations, the black area represents the scanner workstation, and the blue areas represent the post-scanner segments. The archive area corresponds to a transitory paraffin block archive. The stars sign the bottleneck areas. The black arrows comprehend physical traffic of samples, and the gray arrows represent traffic of digital information.

**Figure 3 diagnostics-11-02111-f003:**
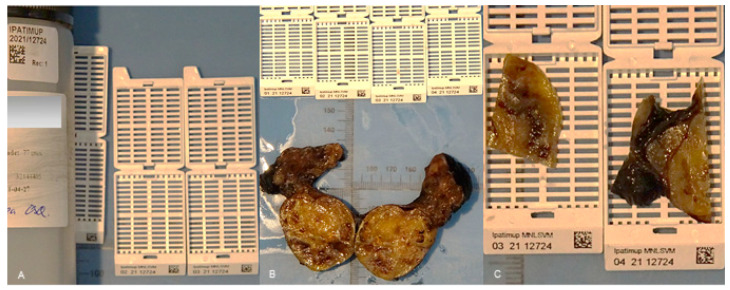
Containers with specimens are labeled with QR codes and photographed at arrival (**A**). Photographic documentation of the specimen is performed (**B**) as well as of the selected fragments inside the respective labeled cassettes (**C**).

**Figure 4 diagnostics-11-02111-f004:**
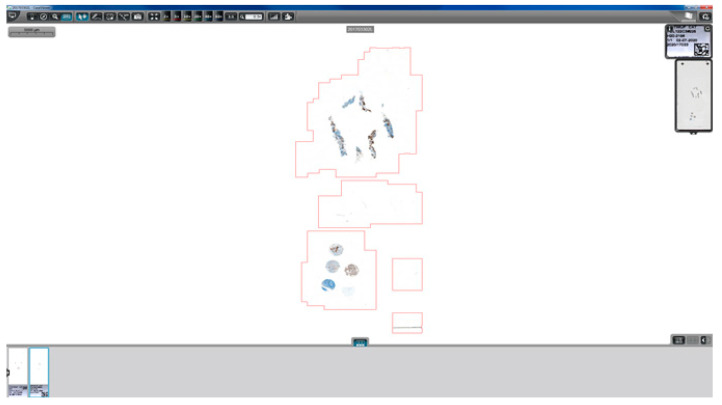
Print screen of a whole slide image representing an immunohistochemistry staining of a specimen together with the set of 5 control tissue microarrays for routine utilization.

**Table 1 diagnostics-11-02111-t001:** WSI bright field results of the 8 months operating fully digitally.

	Month								
	July 2020	August 2020	September 2020	October 2020	November 2020	December 2020	January 2021	February 2021	Mean Value
Slides scanned (*n*)	7047	5818	8159	9099	7807	7135	6349	6004	7177
Cases scanned (*n*)	1688	1335	1814	1871	1697	1307	1290	1361	1545
Cases re-scanned (*n*; %) (by technique order)	30; 1.8	23; 1.7	31; 1.7	27; 1.4	5; 0.3	1; 0.1	5; 0.4	4; 0.3	16; 1.0
Cases with good image (%) (by pathologist order)	96.3	97.6	99.0	98.9	98.5	99.1	98.5	98.5	98.3
Cases with glass slides requested (%)	2.1	1.6	1.6	2.1	2.0	2.2	3.3	3.8	2.3

**Table 2 diagnostics-11-02111-t002:** Average sizes of slides and respective times for scanning concerning the type of specimen.

Type of Preparation		H&E	Histochemistry	Immunohistochemistry	Bright Field In Situ Hybridization
Type of Sample					
Small biopsy	Mean size (megabytes)	242	203	266	4767
	Mean time (seconds)	48	43	52	211
Large specimen	Mean size (megabytes)	1625	2046	1496	9930
	Mean time (seconds)	109	151	101	392

## Data Availability

All data generated or analyzed during this study are included in this published article.

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
