# Peer review of "Digital Pathology Workflow Implementation at IPATIMUP"

_diagnostics, 2021, doi:10.3390/diagnostics11112111_

Round 1

Reviewer 1 Report

This manuscript under the title “Digital Pathology Workflow Implementation at IPATIMUP” describes the process of digital transformation of the pathology laboratory at IPATIMUP, including a detailed description of the modifications operated throughout the processing pipeline, as well as the advantages of its implementation.

The entire manuscript is properly written but it could be substantially improved if schematic representation illustrating the pre and post scanner steps of the digital workflow in the corresponding facility, together with major advantages, disadvantages, and bottlenecks of the entire process are presented.

Thus the minor revision of the manuscript is recommended.

Author Response

Dear Reviewer 1, thank you for considering this paper for publication. here you have the reply to your comments.

The entire manuscript is properly written but it could be substantially improved if schematic representation illustrating the pre and post scanner steps of the digital workflow in the corresponding facility, together with major advantages, disadvantages, and bottlenecks of the entire process are presented.

A figure 2 with detailed legend and an explanatory note was introduced in the text, documenting the workflow in the lab.

Thank you.

Best regards,

Reviewer 2 Report

This study describes the digital transformation process of the Institute of Molecular Pathology and Immunology of University of Porto (IPATIMUP) in Porto, Portugal. The advantages of the implanted digital model are highlighted. However, my main doubt is whether this descriptive study meets the purposes of the journal. The manuscript contains four keywords, three figures, two tables, and fourteen references. In general, it is a correct manuscript, although some remarks are made.

Please, avoid the use of not well-known abbreviations in the title.
For keywords, where possible, please use Medical Subject Headings terms (MeSH Terms). Initially, only one keyword “telepathology” is a MeSH term.
As you very well do with the rest of the abbreviations in the text, the abbreviation IPATIMUP should be explained the first time it is used.
The manuscript has 14 references. Reference format is according to the journal’s guidelines.

Author Response

Dear Reviewer 2, thank you for considering this work for publication. here you have the reply to your comments.

Please, avoid the use of not well-known abbreviations in the title.

IPATIMUP is not just an abbreviation, is a registered name per se and is well know as it is. This justifies the usage of the IPATIMUP in the title in the current form.

For keywords, where possible, please use Medical Subject Headings terms (MeSH Terms). Initially, only one keyword “telepathology” is a MeSH term.

We changed accordingly. We only kept Digital Pathology keyword, even not being part of the MesH list, because is very relevant to the topic.

As you very well do with the rest of the abbreviations in the text, the abbreviation IPATIMUP should be explained the first time it is used.

We performed as suggested.

Best regards,